# Non-canonical activation of the ER stress sensor ATF6 by *Legionella pneumophila* effectors

Nnejiuwa U Ibe[1,2,*], Advait Subramanian[1,2,3,*], Shaeri Mukherjee[1,2]

**The intracellular bacterial pathogen *Legionella pneumophila* (*L.p.*) secretes ~330 effector proteins into the host cell to sculpt an ER-derived replicative niche. We previously reported five *L.p.* effectors that inhibit IRE1, a key sensor of the homeostatic unfolded protein response (UPR) pathway. In this study, we discovered a subset of *L.p.* toxins that selectively activate the UPR sensor ATF6, resulting in its cleavage, nuclear translocation, and target gene transcription. In a deviation from the conventional model, this *L.p.*–dependent activation of ATF6 does not require its transport to the Golgi or its cleavage by the S1P/S2P proteases. We believe that our findings highlight the unique regulatory control that *L.p.* exerts upon the three UPR sensors and expand the repertoire of bacterial proteins that selectively perturb host homeostatic pathways.**

## Introduction

Several intracellular pathogens, including *Legionella pneumophila* (*L.p.*), expertly manipulate host cell function to create their replicative niche. *L.p.* uses the specialized Dot/Icm Type IVB secretion system (T4SS) to translocate roughly 300 bacterial effector proteins into the host cytosol (Berger & Isberg, 1993; Vogel et al, 1998; Isberg et al, 2009; Hubber & Roy, 2010). Once deposited into the cytosol, the effectors target a vast array of host proteins and can influence diverse biological processes which permit the use of *L.p.* as a tool to uncover novel biological mechanisms. During infection, *L.p.* uses its effectors to prevent fusion of the *Legionella*-containing vacuole (LCV) with the host endosomal machinery. Instead these effectors facilitate the remodeling of the LCV into a compartment that supports pathogen replication (Marra et al, 1992; Roy et al, 1998; Wiater et al, 1998). Although fusion with lysosomes is evaded during infection, there is substantial interaction between the LCV and other host organelles including the ER (Horwitz & Silverstein, 1983; Swanson & Isberg, 1995; Tilney et al, 2001). The ER–LCV interactions take on different forms as LCV maturation progresses.

Early during infection, *L.p.* induces tubular ER rearrangements and intercepts ER-derived vesicles destined for the Golgi (Kagan & Roy, 2002; Kotewicz et al, 2017). However, at later time points post infection, the mature LCV is substantially different, becoming studded with ribosomes and reticular ER proteins (Roy & Tilney, 2002), thus highlighting the complexity of interactions between the ER and LCV.

The ER serves as a critical regulatory site for protein and membrane lipid biosynthesis, and imbalances in protein load or membrane lipid perturbations can disrupt many of its vital homeostatic functions (Rapoport, 2007; Fagone & Jackowski, 2009). The unfolded protein response (UPR) serves as a prominent regulatory pathway that has been shown to respond to the burden of accumulating unfolded or misfolded proteins in the ER (Ron & Walter, 2007). In mammalian cells, the UPR is coordinated by three ER-localized transmembrane proteins, inositol-requiring protein-1 (IRE1), protein kinase RNA (PKR)-like ER kinase (PERK), and activating transcription factor-6 (ATF6), each of which initiate pathways designed to modulate the cellular response (Cox et al, 1993; Mori et al, 1993; Harding et al, 1999; Haze et al, 1999).

ATF6 is a type II transmembrane protein that is retained in the ER under normal homeostatic conditions through interactions with the resident chaperone BiP/GRP78 (Shen et al, 2002). Upon accumulation of unfolded proteins, the ER stress stimulates ATF6 translocation from the ER to the Golgi. At the Golgi, ATF6 is sequentially cleaved first by site-1 protease (S1P) enzyme in the lumenal domain, and then by site-2 protease (S2P) enzyme, liberating the cytosolic ATF6-N terminal fragment, ATF6-N (Ye et al, 2000; Shen & Prywes, 2004). Once cleaved, ATF6-N is recruited to the nucleus where it binds to cis-acting ER stress response elements (ERSE) in the promoter regions of UPR target genes (Yoshida et al, 2000; Kokame et al, 2001). ATF6 activation is thought to facilitate cytoprotective adaptation to ER stress through the regulation of genes that improve protein folding and processing in the ER. ATF6 has been shown to suppress the UPR-induced apoptotic program once the stress is resolved (Wu et al, 2007), highlighting the pro-survival contributions of this signaling network.

Studies emphasizing cross talk between the UPR and bacterial infection have revealed an interconnectedness of ER stress sensing

[1]Department of Microbiology and Immunology, University of California, San Francisco, San Francisco, CA, USA   [2]George Williams Hooper Foundation, University of California, San Francisco, San Francisco, CA, USA   [3]Department of Biochemistry and Biophysics, University of California, San Francisco, San Francisco, CA, USA

Correspondence: shaeri.mukherjee@ucsf.edu
*Nnejiuwa U Ibe and Advait Subramanian contributed equally to this work

and pathogen-sensing mechanisms in the cell (Celli & Tsolis, 2015). Pathogenic perturbations endured during infection can impact ER homeostasis in a manner that can also induce ER stress responses. Intracellular pathogens across all kingdoms, from virus to protozoans, have devised strategies to subvert or use one or more UPR programs to benefit survival and replication within the host (Celli & Tsolis, 2015; Galluzzi et al, 2017). As further evidence, studies have demonstrated pathogen-mediated targeting of ATF6 can be beneficial for survival (Ambrose & Mackenzie, 2013; Hou et al, 2017, 2019) and replication (Yoshikawa et al, 2020).

Our previous analysis on *L.p.*–mediated manipulation of the UPR revealed a dynamic processing of full length ATF6 protein levels during infection (Treacy-Abarca & Mukherjee, 2015). To further understand the relationship between *L.p.* infection and ATF6 processing, we sought to understand the mechanism by which *L.p.* modulates the ATF6 pathway. Here, we present evidence of a unique, non-canonical mode of ATF6 activation by *L.p.* that is effector driven and does not rely on host proteins that were previously thought to be essential for ATF6 processing and activation. Interestingly, we discover novel *L.p.* effectors that play a role in the activation of ATF6 during infection.

# Results

## *L.p.* activates the recruitment and processing of ATF6 in an effector dependent manner

In cells subjected to ER stress with the strong reducing agent DTT, full length ATF6 (ATF6-FL) is processed upon cleavage into an ~55 kD N-terminal fragment (ATF6-N) that translocates to the nucleus and activates transcription (Ye et al, 2000; Chen et al, 2002). Both the loss of ATF6-FL and the accumulation of ATF6-N are readily observed by immunoblotting with antibodies raised against ATF6 (DTT lane, Fig 1A). We monitored ATF6-FL processing during *L.p.* infection using HEK293 cells stably expressing the Fcγ receptor (HEK293-FcγR) to allow for the antibody-mediated opsonization of *L.p.* Surprisingly, and in contrast to DTT treatment, infecting these cells with wild type *L.p.* (*WT L.p.*) resulted in the near complete processing of endogenous ATF6-FL into two distinct fragments—a major fragment of ~75 kD that we designate as ATF6-P (*WT L.p.* lane, Fig 1A) and a minor fragment of ~30 kD that we designate as ATF6-LMW (*WT L.p.* lane, Fig 1A, see high exposure). The magnitude of ATF6-FL processing induced by *WT L.p.* was similar to that induced by DTT treatment (Fig 1A). Importantly, infecting cells with an isogenic strain of *L.p.* that lacks a functional secretion system (Δ*dotA L.p.*) did not affect ATF6-FL protein levels (Δ*dotA L.p.* lane, Fig 1A), suggesting that one or more secreted bacterial effectors activates its cleavage in cells. As infection progresses, the LCV is remodeled from a plasma membrane derived vacuole to an ER-like compartment in a process that involves the recruitment of host ER proteins to the LCV and the disruption of ER-to-Golgi trafficking (Swanson & Isberg, 1995; Tilney et al, 2001). Upon monitoring the localization of an N-terminal GFP tagged ATF6 fusion protein (GFP-ATF6-FL, see Fig S1A) in FcγR expressing Cos7 cells by confocal microscopy, we observed a substantial recruitment of ATF6 to the LCV membrane with over 80% of LCVs marked positive for ATF6 (Fig S1B). A majority of LCVs were marked positive for ATF6 in

cells infected with *WT L.p.*, and the recruitment required a functional dot/Icm system as Δ*dotA L.p.* infected cells exhibited significantly lower ATF6 recruitment.

## Proteasome-dependent degradation pathways are not required for ATF6 loss during *L.p.* infection

The processing of ATF6 through regulated intramembrane proteolysis catalyzed by the S1P and S2P proteases has been studied extensively (Ye et al, 2000; Okada et al, 2003), yet degradative processing events have also been shown to control ATF6 levels even in the absence of ER stress (Hong et al, 2004; Horimoto et al, 2013). Interestingly, protein synthesis attenuation during *L.p.* infection has been shown to influence the IRE-1 branch of the UPR (Hempstead & Isberg, 2015; Treacy-Abarca & Mukherjee, 2015); but its impact on ATF6 had not been elucidated.

To test if proteasomal degradation contributed to the observed loss of ATF6-FL, we first monitored ATF6 processing in the presence or absence of proteasome inhibition, under conditions of protein synthesis arrest using the drug cycloheximide (CHX). HEK293-FcγR cells were pre-treated for 3 h with the proteasome inhibitor MG-132 or control media. ER stress induction using DTT led to rapid ATF6 processing after 1 h, whereas prolonged exposure to DTT for 3 h resulted in recovery of ATF6 signal due to autoregulatory feedback from UPR induction (Fig S1C). In contrast to UPR induction, cells treated with CHX showed a loss of the ATF6-FL signal 3 h after treatment (Fig S1C). Whereas CHX treatment alone resulted in reduced levels of ATF6-FL, pre-treatment with MG-132 stabilized the protein in the presence of CHX to pre-treatment levels (Fig S1C), consistent with previously described observations (Haze et al, 1999). We then tested whether proteasomal inhibition could stabilize ATF6-FL protein levels in cells infected with the *WT L.p.* or the Δ*dotA L.p.* strains. Similar to UPR induction using DTT, MG-132 treatment did not protect ATF6-FL processing during *WT L.p.* infection (Fig 1B). When cells were infected with Δ*dotA L.p.*, ATF6-FL remained at pre-infection levels and treatment with MG-132 did not have a significant impact (Fig 1B). Previous studies have identified ATF6 as an ER-associated degradation (ERAD) substrate that undergoes constitutive degradation mediated by SEL1L (Horimoto et al, 2013). It was shown that ATF6 is a short-lived protein with a half-life of less than 2 h and the stability of ATF6 can be markedly increased by *SEL1L* depletion (Horimoto et al, 2013). To test if SEL1L dependent ERAD contributes to loss of ATF6-FL during infection, we next compared ATF6 processing in HEK293-FcγR cells that were treated with non-targeting or *SEL1L*-targeting siRNA. SEL1L knock-down led to an increase in ATF6-FL levels by 1.5-fold in samples not treated with CHX (Fig 1C). Similarly, while CHX treatment for 2 h caused a reduction in ATF6-FL levels in cells transfected with non-targeting siRNA, ATF6-FL levels were again increased by 1.5-fold in *SEL1L* depleted cells (Fig 1C). However, there was no significant change observed in the ATF6-FL signal intensity under *L.p.* infection when normalized to loading controls (Fig 1C). As *L.p.* infection causes an inhibition of protein synthesis (Belyi et al, 2006; Fontana et al, 2011; Tzivelekidis et al, 2011), we considered this deregulation might contribute to the processing of ATF6 during infection. To evaluate the impact of protein synthesis inhibition, we assayed for ATF6 processing in cells infected with a mutant *L.p.* strain Δ*7-Trans-L.p.*

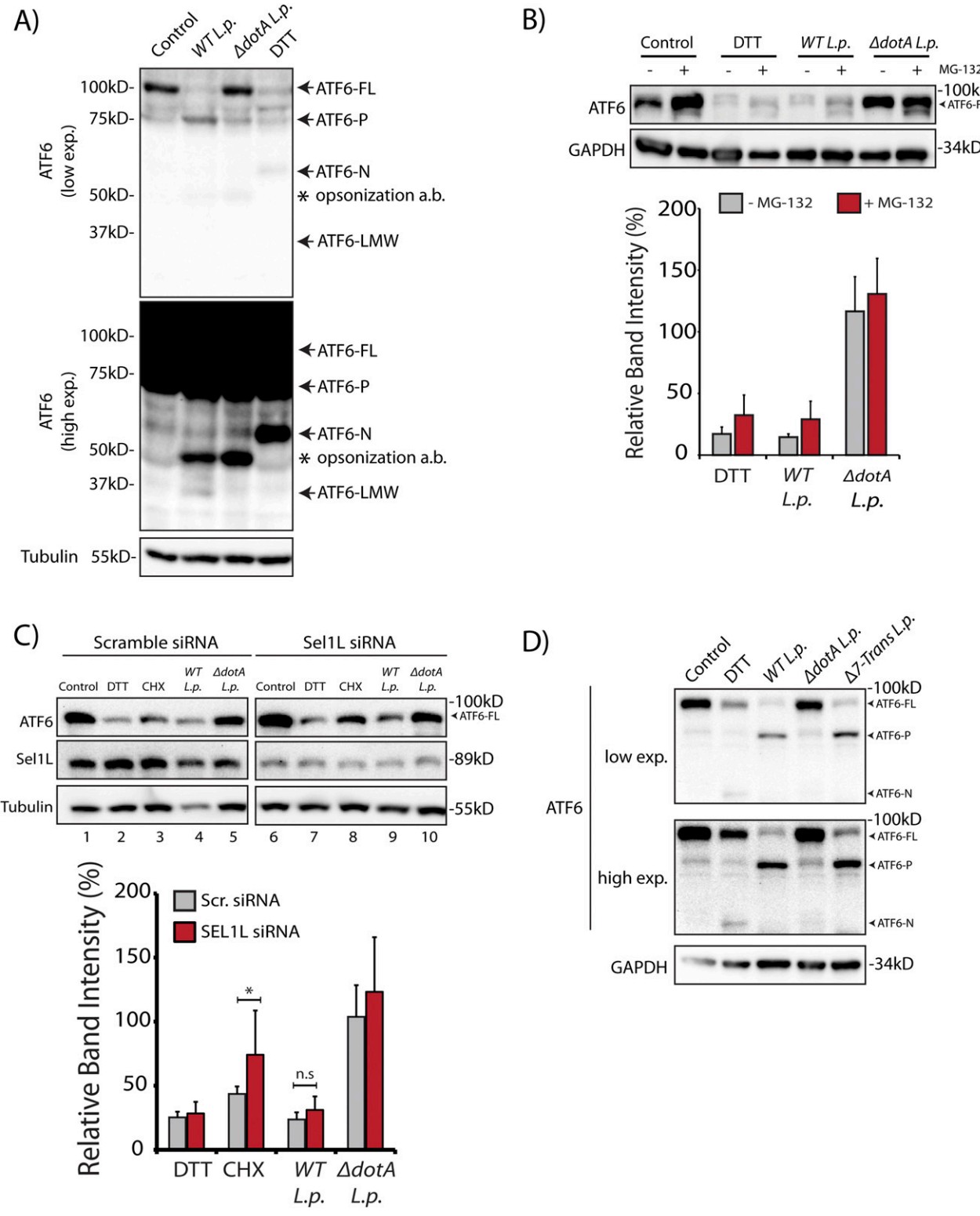

**Figure 1. L. p. infection induces ATF6 processing independently of proteasomal degradation and ERAD.**
**(A)** HEK293-FcγR cells were treated with 1 mM DTT for 1 h or infected with *Legionella* pneumophila (*WT* or Δ*dotA*) for 6 h before harvesting. Lysates were subjected to immunoblotting analysis using the anti-ATF6 and anti-Tubulin antibodies. Low exposure (low exp.) and high exposure (high exp.) blots are shown for ATF6. Arrows mark the precursor full-length ATF6 (ATF6-FL) fragment and the processed ATF6-P, ATF6-N, and ATF6 low molecular weight (LMW) fragments; * marks the opsonization antibody. **(B)** HEK293-FcγR cells were pre-treated with 20 μM MG-132 for 3 h before indicated treatment conditions. **(B)** Cells were treated with 1 mM DTT for 1 h or infected with *WT* or Δ*dotA L.p.* for 6 h (MOI = 5) and analyzed by immunoblotting using anti-ATF6 and anti-GAPDH antibodies. Histograms represent quantitation of ATF6-FL signal from

that lacks seven characterized T4SS effectors that are known to block protein synthesis (Fontana et al, 2011; Barry et al, 2013). Consistent with our previous results, we observed ATF6 cleavage and processing to the ATF6-P fragment at similar levels when cells were infected with either the Δ7-Trans-L.p. strain or WT L.p. (Fig 1D). Taken together, these experiments suggest ATF6-FL processing during L.p. infection is not a direct result of enhanced proteasomal degradation or a consequence of protein synthesis arrest.

### L. p. induces ATF6-mediated gene induction via the ATF6-LMW fragment

We next considered how the L.p. induced ATF6-FL processing into the ATF6-P (~75 kD) and ATF6-LMW (~30 kD) fragments affected its distal function as a nuclear transcription factor. To address this question, we used the N-terminal tagged GFP fusion protein of ATF6-FL (GFP-ATF6-FL) (Fig S1A) (Chen et al, 2002). During ER stress, GFP-ATF6-FL traffics from the ER to the Golgi where it is cleaved by S1P and S2P proteases to release the ~80 kD soluble N-terminal ATF6 fragment (GFP-ATF6-N) that translocates to the nucleus and activates transcription (Chen et al, 2002; Nadanaka et al, 2004) Indeed, in HeLa cells stably expressing the Fcγ receptor (HeLa-FcγR) and co-transfected with GFP-ATF6-FL and a RFP tagged fusion of the enzyme galactosyl transferase (GalT-RFP) to mark the Golgi, DTT treatment caused a rapid translocation of GFP-ATF6-FL from the ER to the Golgi within 30 min, where the GFP signal colocalized with the Golgi marker GalT-RFP (Fig S2A and Videos 1). After 120 min of DTT treatment, the GFP signal predominantly localized to the nucleus, corresponding to an accumulation of the nuclear GFP-ATF6-N fragment (Fig S2A). We then monitored the dynamics of GFP-ATF6-FL in cells infected with Halo-tagged L.p. and stained with the cell permeable dye JF-644 (Fig 2A and Videos 2). Analyses of time-lapse micrographs of cells at 1 and 6 h post infection revealed a significant increase in the nuclear signal intensity of GFP-ATF6 only during WT L.p. infection at the latter time point (Fig 2A and B). Strikingly, GFP-ATF6 did not seem to be recruited to the Golgi/perinuclear region during L.p. infection, as seen with DTT treatment (Videos 2). These results suggested that L.p. infection induces the cleavage of GFP-ATF6-FL into a fragment that remains fused to GFP and enters the nucleus. To determine the identity of this fragment, we generated four N-terminal GFP tagged truncation mutants of the cytosolic domain of ATF6 of different lengths—GFP-ATF6 1-291 lacked the basic leucine zipper (bZIP) domain, whereas GFP-ATF6 1-331, GFP-ATF6 1-343, and GFP-ATF6 1-355 possessed partial bZIP domains (Fig 2C). We collected lysates from cells transfected with these GFP-ATF6 N-terminal truncation mutants (Fig 2D; first four lanes) and compared and contrasted their relative molecular weights (MWs) with that of the GFP-ATF6 LMW fragment induced by WT L.p. (Fig 2D). We observed a loss of GFP-ATF6-FL, as expected, and a concomitant and robust accumulation of

a fragment of ATF6 fused to GFP at a MW of ~60 kD (GFP-ATF6-LMW) 5 h post infection with WT L.p. but not ΔdotA L.p. (Fig 2D). We determined that the fragment induced by WT L.p. migrated on a reducing SDS–PAGE gel in a manner similar to the GFP-ATF6 1-331 truncation mutant (Fig 2D, compare 1–331 lane with WT L.p. 5 h lane at high exposure). Significantly, an in vitro transcription and translation product of the DNA encoding the ATF6 1-331 protein migrated as a sharp band at a MW of ~30 kD on a reducing SDS–PAGE and was detected by an antibody raised against ATF6 (Fig 2E). This ~30 kD size is consistent with the endogenous ATF6-LMW fragment generated during WT L.p. infection (Fig 1A). Taken together with the observations that the ATF6 1–331 fragment accumulates in the nucleus (Yoshida et al, 2001), these findings suggested to us that the ATF6-LMW generated by WT L.p. carries a partial bZIP domain and translocates to the nucleus.

To evaluate transcriptional activation during L.p. infection, we monitored the mRNA levels of ATF6 target genes by quantitative real time PCR (qRT-PCR), including UPR regulator/ER chaperone BiP (HSPA5), in HEK293-FcγR cells. As expected, UPR induction with DTT increased expression of ER quality control genes BiP and HERPUD1 by greater than fivefold in comparison to control (DMSO-treated) cells (Fig 3A). Interestingly, these ATF6-regulated genes were also induced in WT L.p. infected cells by greater than fivefold. As an added control, when compared with cells treated with protein synthesis inhibitor CHX, we observed that CHX had no effect on BiP gene expression (Fig 3B). To gain more insight into ATF6 mediated transcript induction patterns during L.p. infection, we examined the gene activation profile of BiP at different time points over the course of an infection. In ER-stressed cells, analysis of BiP mRNA indicated a spike in expression between 4 and 5 h post DTT treatment (Fig 3C). When the expression profile was examined under avirulent ΔdotA L.p. infection conditions, BiP expression spiked early on, 1 h post-infection, but quickly dropped to pre-infection levels at later time points assayed. In contrast, infections with WT L.p. stimulated a robust increase in BiP expression, which continued even up to 7 h post-infection.

We then undertook two orthogonal approaches to determine if the transcriptional program induced during L.p. infection was dependent on the ATF6-LMW. As the ATF6-LMW resembled the sequence architecture of the ATF6 1-331 mutant protein (see Fig 2D and E), we first determined if the ATF6 1–331 fragment was capable of binding the ER stress response element (ERSE) motif in the nucleus and activate transcription. To characterize ERSE binding, we used a previously validated HEK293T cell line comprising of a stably integrated tandem ERSE motif with regulatory control over luciferase expression (HEK-ERSE-Luc) (Gallagher et al, 2016) (Fig 3D). Indeed, expression of the ATF6 1–331 fragment in this reporter cell line significantly activated the expression of luciferase downstream of the ERSE sequence albeit at a level lower than when the full length ATF6 proteins or the ATF6 1–355 and ATF6 1–373 (ATF6-N) fragments were expressed (Fig 3D). The ATF6 1–291 fragment that

---

replicate experiments (n = 3). **(C)** Cell lysates from scramble or SEL1L siRNA transfected into HEK293-FcγR cells were analyzed by immunoblotting using anti-ATF6, anti-SEL1L, and anti-Tubulin antibodies. CHX treatment was performed with 25 μM CHX for 2 h. Histograms represent quantitation of ATF6-FL signal from replicate experiments (n = 3). Mean ± SEM. P-values were calculated using t test. *P < 0.05. **(D)** HEK293-FcγR cells were treated with 1 mM DTT or infected with WT L.p., ΔdotA L.p., and Δ7-translation mutant (Δ7-Trans) L.p. strains as described in (A). Lysates were subjected to immunoblotting analysis using the anti-ATF6 and anti-GAPDH antibodies. Arrows mark the precursor full-length ATF6 (ATF6-FL) fragment and the processed ATF6-P and ATF6-N fragments.

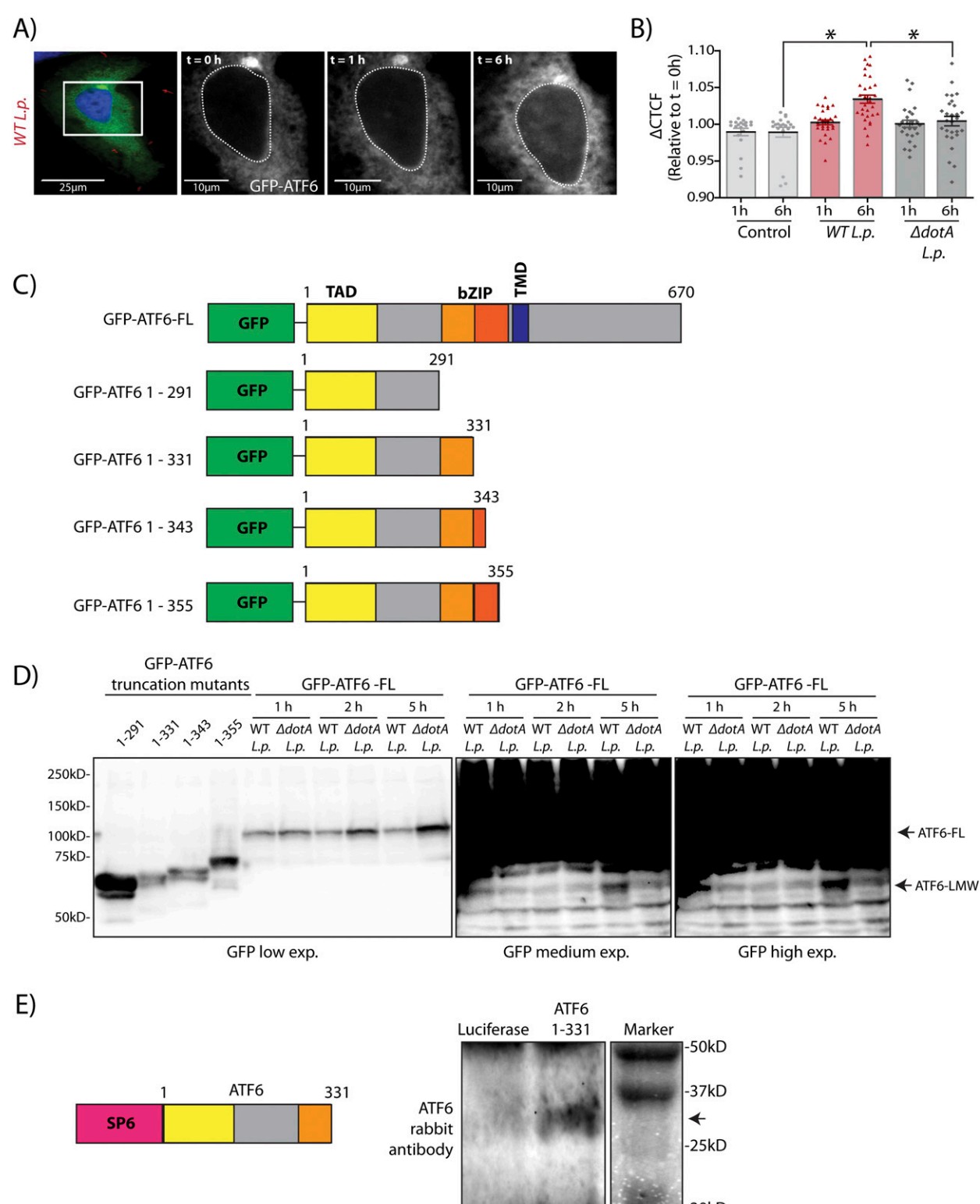

**Figure 2. *L.p.* infection generates a low molecular fragment that carries a partial bZIP domain.**
**(A)** Representative images from the time-lapse wide-field confocal microscopic analyses of HeLa-FcγR cells transiently transfected with GalT-RFP and GFP-ATF6 and then infected with *Halo-tag WT-L.p* pre-stained with JF-646. Live-cell epifluorescence image stills at time point 0 (t = 0) (panel 1, scale bar = 25 μm), with white boxes denoting location of inset. GFP-ATF6 signal at 0, 1, and 6 h post infection (inset panels, scale bars = 10 μm). **(B)** Quantitation of temporal changes in nuclear localized GFP-ATF6 relative to the corrected total cell fluorescence (CTCF) of ATF6 in single cells at 1 and 6 h post infection. Signal intensities analyzed from of control (n = 20), *WT L.p.* (n = 31) or *ΔdotA L.p.* (n = 28) were normalized to CTCF signal at time t = 0. Mean ± SEM. *P*-values were calculated using *t* test. **P* < 0.05. **(C)** Schematic representation of

lacked the bZIP domain failed to induce luciferase expression (Fig 3D). Furthermore, the ATF6 1–331 fragment but not the ATF6 1–291 fragment significantly increased the transcription of ATF6 target genes *BiP* and *DNAJB11* in HEK293T cells (Fig S2B). To complement these findings, we depleted ATF6 during *L.p.* infection and assayed for *HSPA5/BiP* expression in HEK293-FcγR cells. The ATF6 mRNA was targeted for knockdown in HEK293-FcγR cells using siRNA duplexes that achieve a greater than 80% knockdown efficiency (Fig 3E). Indeed, ATF6 depletion under *L.p.* infections markedly reduced *BiP* mRNA induction, suggesting that the cleavage of endogenous ATF6 is necessary to induce downstream gene activation. However, we note here that in comparison with DTT treatment (Fig 3E, black bars), elevated *BiP* mRNA levels in ATF6 depleted cells were still persistent in response to *WT L.p.* infection (Fig 3E, red bars). These results suggest that the ATF6-LMW fragment accounts for a significant fraction of the ATF6 target gene induction in *L.p.* infected cells. However, it does not account for all the transcripts induced. Certain *L.p.* effectors might bypass the requirement for ATF6 cleavage entirely by directly inducing *BiP* transcription, suggestive of a redundant strategy often used by *L.p.* (see the Discussion section).

In sum, the gene expression changes in concomitance with ATF6 processing induced by *L.p.* demonstrate the activation of the ATF6 pathway during *L.p.* infection.

### L. p. induces a non-canonical ATF6 activation

Our results so far revealed two surprising findings: (1) During *L.p.* infection, ATF6-FL is processed into a ATF6-P fragment and a ATF6-LMW fragment that retains transcriptional activity (Figs 1A and 3A–E); and (2) *L.p.* does not stimulate ATF6 translocation to the Golgi apparatus (Videos 2). It is well understood that *L.p.*'s vast effector repertoire permits its subversion of numerous host pathways (Omotade & Roy, 2019). However, it remains unknown which aspects of ATF6 activation during infection are host-driven or effector-driven. To further understand the mechanism of the ATF6 dependent transcriptional program in *L.p.* infected cells, we performed a pharmacological inhibition of host pathways that are thought to be essential for ATF6 activation. First, we used Ceapin-A7, a class of inhibitors that selectively tether ATF6 to the peroxisomal protein ABCD3 and prevent its activation by inhibiting its translocation from the ER to the Golgi (Gallagher et al, 2016; Gallagher & Walter, 2016; Torres et al, 2019). Treatment of both HEK293 cells and RAW264.7 macrophages cells with DTT resulted in the processing of ATF6-FL and an accumulation of ATF6-N (Figs 4A and S3A), leading to a greater than fivefold induction of *BiP* mRNA levels (Figs 4B and S3B). In contrast, co-treatment of cells with DTT and Ceapin-A7 partially protected ATF6 from cleavage (Figs 4A and S3A) and significantly reduced *BiP* expression (Figs 3B and S3B). As shown earlier, *WT L.p.* infection alone leads to the complete processing of ATF6-FL in HEK293 cells (Fig 1A), along with a strong

induction of *BiP* mRNA (see Fig 3). In macrophages, interestingly, *WT L.p.* infection for 1 h resulted in the accumulation of the ATF6-P, ATF6-N and the ATF6-LMW fragments (Fig S3A). Yet surprisingly, pre-treatment of both cell types with Ceapin-A7 did not prevent the accumulation of processed ATF6 fragments that were induced by *WT L.p.* (Figs 4A and S3A). In addition, ATF6 target gene expression levels, including *BiP* mRNA, remained high and persisted at levels similar to infected cells without Ceapin-A7 treatment (Figs 4B and S3B). Neither DTT treatment nor *WT L.p.* infection elicited any significant changes in the expression of the Ceapin induced ATF6 tethering partner protein ABCD3 (Fig 4A). Cumulatively, these results strongly suggest that ATF6 translocation from the ER to the Golgi is not a pre-requisite step required for its processing during *L.p.* infection.

*L.p.* is known to exploit ER-to-Golgi trafficking and individual effectors have been identified that can disrupt Golgi homeostasis (Mukherjee et al, 2011). As disrupted homeostasis could result in mis-localization of Golgi proteins, the impact of the normally Golgi resident proteases S1P and S2P on ATF6 activation were tested. First, we directly inhibited the activity of S1P using the inhibitor PF-429242 (S1Pi) (Lebeau et al, 2018). Inhibition of S1P proteolysis activity in the presence of DTT resulted in the appearance of a slower migrating species of ATF6 at a higher MW likely due to extensive glycosylation in the Golgi (Fig 4C, band marked "&"). Further validating S1P inhibition, *BiP* mRNA induction was reduced by nearly 80% when compared with DTT treatment alone (Fig 4B; +S1Pi bars). Remarkably, S1P inhibition did not alter ATF6 processing and *BiP* mRNA induction in *WT L.p.* infected cells (Fig 4B and C). The cleavage of ATF6 during *L.p.* infection even in the presence of Ceapin-A7 and S1P inhibition are suggestive of an alternative proteolytic mechanism induced by *L.p.*. We next addressed whether the canonical S1P and S2P cleavage sites on ATF6 were a prerequisite for the *L.p.* induced processing of ATF6 to the transcriptionally active ATF6-LMW fragment. To test this, we generated N-terminal GFP tagged ATF6-FL constructs harboring point mutations on only the ATF6 S1P cleavage site (R415A/R416A; GFP-ATF6 S1P mutant) and on both the ATF6 S1P and S2P cleavage sites (R415A/R416A and N391F/P394L; GFP-ATF6 S1P/S2P mutant) (see Fig 4D for a schematic). HEK293-FcγR cells transiently expressing either the GFP-ATF6 S1P mutant or the GFP-ATF6 S1P/S2P mutant were treated with DTT or infected with *WT L.p.* or Δ*dotA L.p.* strains. DTT treatment indicated that the processing of ATF6-FL was greatly impaired in cells expressing the cleavage site mutant constructs compared with wild-type GFP-ATF6 (Fig 4E), with more than 75% of ATF6-FL remaining after DTT treatment. Strikingly, the processing of ATF6-FL and the accumulation of the ATF6-LMW fragment still progressed under *WT L.p.* infection, even in the absence of functional S1P and S2P cleavage sites (Fig 4E, GFP high exposure blots).

Given the findings that *L.p.* induced ATF6-FL processing does not depend on the Golgi localized S1P or S2P activities, we also tested if

N-terminal GFP-tagged ATF6 truncation mutants. TAD, transcription activation domain; bZIP, basic leucine zipper domain; TMD, transmembrane domain.
**(D)** Immunoblotting of lysates from cells transfected with truncation mutants of the ATF6 cytosolic domain or transfected with GFP-ATF6-FL and infected with *L.p.* (*WT* or Δ*dotA*) for 1, 2 and 5 h with an antibody raised against GFP. Cells were pre-treated with 5 μM MG-132 for 1 h and maintained during the infection time course. Arrows mark the GFP-ATF6-FL (low exposure) and the GFP-ATF6-LMW (medium and high exposures). **(E)** Immunoblotting of in vitro transcribed and translated luciferase and the ATF6 1–331 fragment. The ATF6 1–331 fragment was detected using an anti-ATF6 antibody raised in rabbit.

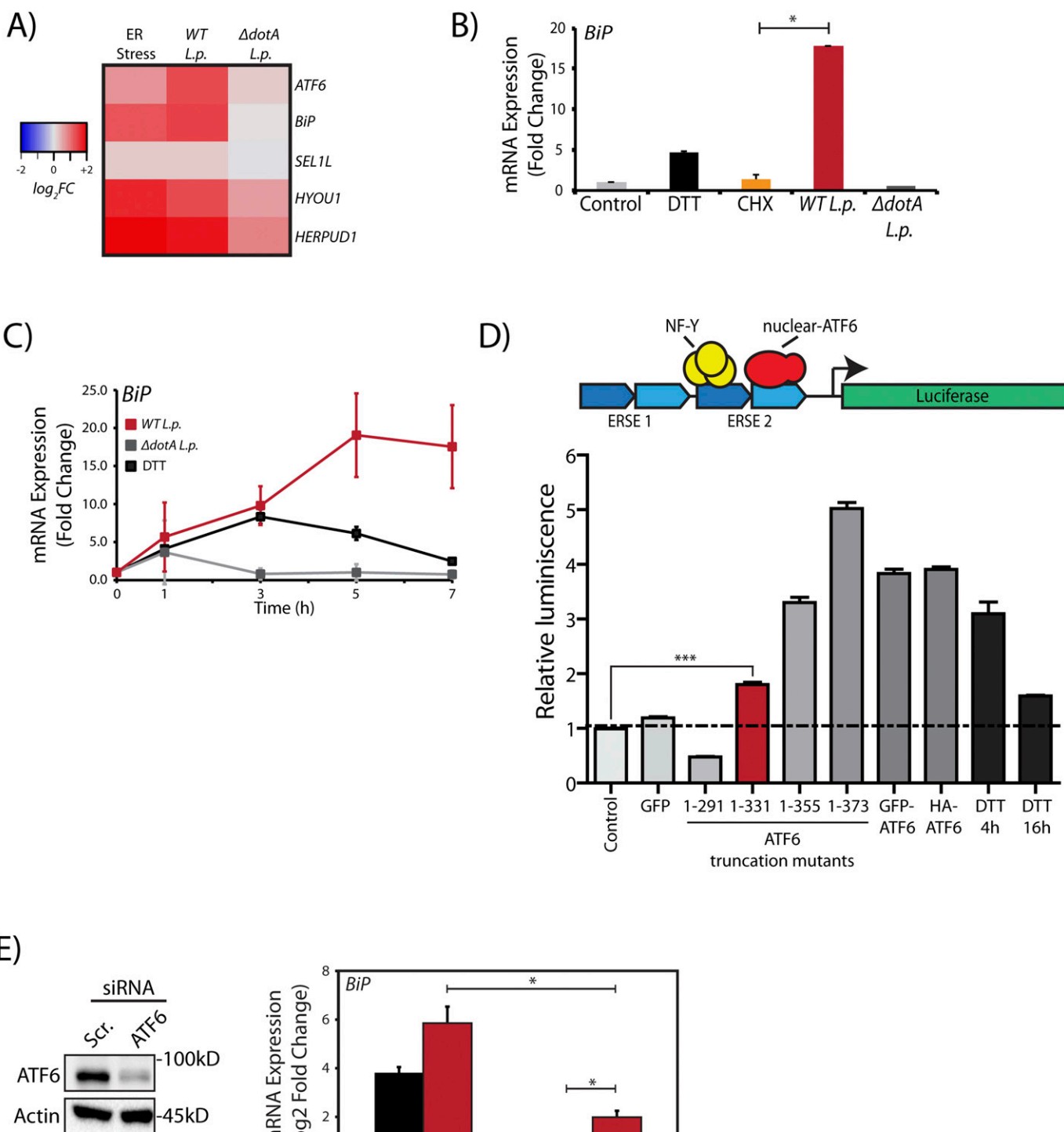

Figure 3. *L.p.* infection induces ATF6 target gene transcription.

**(A)** HEK293-FcγR cells were treated with 1 mM DTT for 6 h or infected with *L.p.* (*WT* or Δ*dotA*) for 6 h before RNA extraction and qRT–PCR analysis monitoring ER quality control genes *ATF6, BiP, SEL1L, HYOU1,* and *HERPUD1*. Heat map depicts log₂ fold changes relative to *GAPDH*. **(B)** Cells were treated with 1 mM DTT or 25 µM CHX for 6 h, or infected with *L.p.* (*WT* or Δ*dotA*) for 6 h. qRT–PCR analysis of *Bip* (*GRP78*) was performed and fold change was calculated relative to GAPDH. **(C)** *L.p.* infected HEK293-FcγR cells (MOI = 100) were harvested post-infection at indicated times and analyzed by qRT–PCR using primers against *Bip*. **(D)** Schematic representation of ERSE-luciferase construct stably expressed in HEK-293T cells (HEK293T-ERSE-Luciferase). Two copies of the ER stress response element (ERSE, blue) are cloned upstream of a minimal promoter driving luciferase (green) gene expression. Once cleaved, the nuclear ATF6 fragment binds the ERSE sequence and stimulates luciferase expression. Histograms

other potential host proteases might regulate this step. To that end, we used a cocktail of five cell permeable protease inhibitors (AEBSF, calpain inhibitor I, E-64 protease inhibitor, PMSF, and TPCK) that have both overlapping and distinct specificities. Pre-treatment of cells with this protease inhibitor cocktail did not affect the loss of ATF6-FL activated by *WT L.p.* (Fig S3C). Furthermore, inhibiting the activity of lysosomal proteases indirectly by treating cells with the vacuolar proton pump inhibitor bafilomycin A1 also had no effect on the processing of ATF6-FL and the accumulation of the ATF6-P and ATF6-LMW fragments induced by *WT L.p.* infection (Fig S3D). Whereas we cannot completely rule out the possibility of an *L.p.* protease that is resistant to the protease inhibitors used, our data cumulatively suggest that *L.p.* infection stimulates the ATF6 pathway in HEK293 cells and mouse macrophages through a mechanism that circumvents the requirement of canonical pathway components.

### ATF6 activation by *L.p.* is strain and species specific

The data presented so far highlights a T4SS-dependent activation strategy requiring the translocation of *Legionella* effector proteins. Therefore, we sought to identify effector proteins capable of inducing ATF6 activation by identifying *Legionella* strains that fail to efficiently process ATF6-FL. Genomic analysis of over 30 *Legionella* strains and species revealed largely non-overlapping effector repertoires (Burstein et al, 2016); therefore, we sought to use a comparative approach to test for ATF6-FL processing in different *Legionella* strains and species. Four *Legionella* species—*L. pneumophila*, *Legionella micdadei* (*L. mic*), *Legionella wadsworthii* (*L. wad*), and *Legionella longbeachae* (*L. lon*)—were tested in addition to *L. pneumophila* strains—*Philadelphia* str. (*WT L.p. Phila* or *ΔdotA L.p. Phila*), *Paris* str. (*L.p. Paris*), *Lens* str. (*L.p. Lens*), and *Serogroup 6* str. (*L.p. SG6*). The *Legionella* species and strains were used to infect HEK293-FcγR and RAW264.7 macrophage cells and endogenous ATF6-FL levels were monitored by immunoblotting. Whereas most of the species and strains tested recapitulated the loss of ATF6-FL as seen with wild type *L.p.* (see above), infecting cells with either *L. wadsworthii* or the *L. pneumophila Paris* str. did not result in an efficient processing of ATF6-FL (Fig S4A and B). Indeed, whereas the WT *L.p. Phila* str. strongly induced the accumulation of the processed ATF6-P fragment in infected cells, this processing was entirely absent in cells infected with the *L. pneumophila Paris* str. (Fig 5A). The ineffectiveness of the *L.p. Paris* str. to process ATF6-FL was not due to a lack of infectivity as cells infected with either the *L.p. Paris* strain or the *L.p. Phila* strain exhibited a robust recruitment of ubiquitin-modified substrates around the LCV (Fig S4C), a hallmark of establishing a successful infectious paradigm (Horenkamp et al, 2014).

Both the *L.p. Phila* strain and the *L.p. Paris* strain evolved from the same species *pneumophila* and thus possess highly similar effector repertoires that are secreted into host cells (Burstein et al, 2016). However, a comparative genomic analysis between the *Philadelphia* str. and *Paris* str. revealed genes that encoded for 17

known *L.p. Philadelphia* str. effector proteins that were absent from *L.p. Paris* str. effector repertoire (Fig 5B). We supposed that one or more of these effectors unique to the *L.p. Phila* strain might regulate the processing of ATF6-FL. To identify individual effectors capable of activating the ATF6 pathway, we used the HEK-ERSE-Luc cell line and screened 16 of these effectors for their ability to activate ERSE dependent transcriptional activation of luciferase (see Fig 3D for a schematic). Each unique effector was epitope-tagged and transiently expressed into the HEK-ERSE-Luc cell line. As controls we included cells treated with ER stress activator thapsigargin (Tg) and cells expressing GFP-ATF6-FL that activates the ERSE reporter (Ye et al, 2000) (Fig 3D). Tg treatment produced a greater than sixfold induction of luciferase activity (Fig 5C, dark red bar), whereas expression of the GFP-ATF6-FL protein led to a fourfold increase in luciferase activity over the control vector, as expected (Fig 5C, pink bar). As a further control, expression of the GFP-ATF6 S1P/S2P cleavage resistant mutant did not lead to an increase in luciferase activity, excluding the possibility that simply overexpressing an ER resident protein triggers ERSE reporter activation (Fig 5C, orange bar). Upon assaying for ERSE reporter activation mediated by the *L.p. Phila* str. effector subset, our experiments revealed that the expression of only five effectors namely *lpg2131*, *lpg0519*, *lpg1948*, *lpg2523*, and *lpg2525* robustly increased luciferase activity by up to threefold (Fig 5C, dark grey bars) as compared with controls, similar in magnitude to the ERSE reporter activation induced by the expression of the ATF6 1-331 fragment (see Fig 3D). Together, these results indicate that multiple *L.p.* effectors possess the capacity to activate transcriptional targets of the ATF6 pathway. We note here that most of the effectors identified in this screen had little or no known functions assigned to them previously.

### Lpg0519 localizes to the ER and activates ATF6

To experimentally validate the results from the screen, we selected the top ranked *L.p.* effectors *lpg0519 and lpg2131* for further characterization. We generated GFP tagged fusion constructs of *lpg0159* and *lpg2131* and transiently expressed them in U2OS and HEK293 FcγR cells. Unfortunately, the ectopic expression of tagged Lpg2131 in cells resulted in toxicity and confounded our interpretation of the results obtained (data not shown). We thus excluded this effector from further analyses. In contrast, GFP-Lpg0519, when expressed in U2OS cells, was relatively non-toxic and localized to the ER as observed by colocalization with the ER marker mCherry-KDEL (Fig 5D). Furthermore, augmenting Myc-Lpg0519 expression in HEK293 FcγR cells resulted in a robust processing of ATF6-FL and an accumulation of the ATF6-P fragment (Fig 5E), similar to observations made with live *L.p.* infections of cells. Lpg0519 overexpression also resulted in an induction of BiP protein levels (Fig 5F). Importantly, the processing of ATF6-FL induced by Lpg0519 continued even in the presence of Ceapin-A7, further indicating that this *L.p.* effector processes ATF6-FL in a non-canonical manner (Fig 5G). Our

---

depict luminescence units relative to control from the ERSE-Luciferase reporter cell line (n = 3). HEK293T-ERSE-Luciferase cells were transiently transfected with the constructs as indicated for 48 h. Luminescence units were internally normalized to non-transfected control cells. Mean ± SEM. *P*-values were calculated using *t* test. ***P < 0.001. **(E)** *ATF6* siRNA or scramble siRNA transfected into HEK293-FcγR cells were processed and analyzed by immunoblotting using anti-ATF6 and anti-Actin antibodies. Transcript levels were analyzed by qRT-PCR using primers against *BiP* (n = 4). *P < 0.05.

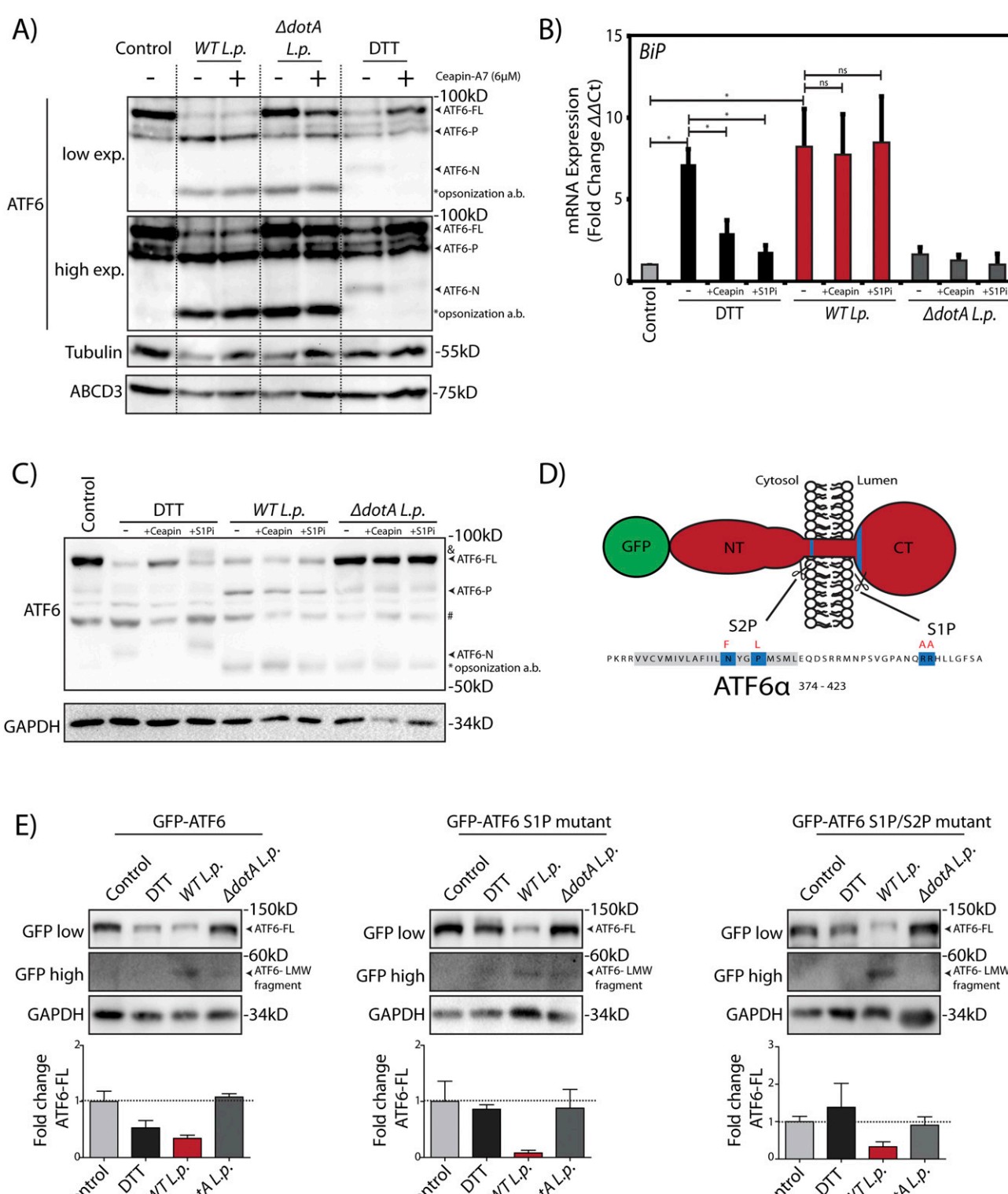

**Figure 4. L.p. induces non-canonical activation of ATF6.**

**(A)** HEK293-FcγR cells were treated 6 µM Ceapin A7 (+Ceapin) for 1 h prior to infection or 1 mM DTT treatment for 1 h. Cells were infected with L. p. (WT or ΔdotA) for 6 h, then lysed and analyzed by immunoblotting using anti-ABCD3, anti-ATF6 and anti-Tubulin antibodies. Arrows mark the ATF6-FL or processed ATF6-P or ATF6-N fragments; "*" marks the opsonization antibody (a.b.) used to coat L.p. before infection. **(B, C)** HEK293-FcγR cells were treated 6 µM Ceapin A7 (+Ceapin) or 1 mM PF-429242 (+S1Pi) for 1 h before infection or DTT treatment. **(B, C)** DTT treated samples were incubated with 1 mM DTT for (C) 1 h or (B) 4 h. **(B, C)** Cells were infected with L. p. (WT or ΔdotA) for 6 h, then lysed and (C) analyzed by immunoblotting using anti-ATF6 and anti-GAPDH antibodies and (B) analyzed by qRT-PCR using primers against BiP. In figure (C), band

experiments also revealed that while ER stress activation with Tg resulted in an increase of both BiP and ATF4 protein levels in cells, Lpg0519 expression did not induce ATF4 (Fig 5F). These results highlight the specificity of Lpg0519 in activating the ATF6 pathway without inducing other signaling arms of the UPR. Altogether, these data suggest that Lpg0519 localizes to the ER and has the capacity to specifically induce the cytoprotective branch of UPR (ATF6), without affecting the pro-apoptotic branch (PERK).

# Discussion

The main finding of this study is that *L. pneumophila* secretes toxins that specifically induce the processing and activation of the ER stress sensor ATF6 in a manner that is both unique and distinct from the ATF6 activation pathway induced in response to the buildup of unfolded proteins in the ER. The salient features of the *L.p.* induced phenomenon that distinguish it from the canonical UPR pathway are as follows: (1) ATF6-FL is processed into two fragments, a higher MW ATF6-P fragment (~75 kD) and a lower MW ATF6-LMW fragment (~30 kD) (Fig 1A). Analysis of the MWs of the respective fragments supported by experiments conducted with the N-terminally tagged GFP-ATF6-FL protein suggest that the ATF6-LMW fragment once cleaved carries a partial bZIP domain, is soluble and enters the nucleus, while the ATF6-P fragment is possibly a membrane associated remnant (Figs 1A and 2). Indeed, the ATF6-LMW fragment is capable of binding the ERSE motif of ATF6 target genes in the nucleus and activates their transcription (Figs 3D and S2B); (2) The generation of the ATF6-P and ATF6-LMW fragments is entirely independent of the established canonical pathway of ATF6 activation that requires its trafficking to the Golgi apparatus and cleavage by S1P and S2P proteases (Figs 4 and S3A–B); and (3) only certain strains of *Legionella spp.* secrete toxins that are capable of processing ATF6 in this non-canonical fashion (Figs 5A and S4 and B). These features highlight novel regulatory and evolutionary aspects of ATF6 biology that we discuss below.

### The impact of ATF6-LMW on transcript induction

As noted earlier, the ATF6-LMW fragment resembles the ATF6 1–331 fragment in molecular architecture and size and contributes towards a significant fraction of the ATF6 mediated transcriptional response in *L.p.* infected cells (Figs 2C–E and 3D). However, its capacity to bind the ERSE sequence and activate transcription is two orders of magnitude lower than the ATF6-N fragment (ATF6 1–373) (Fig 3D), potentially due to the fact that the ATF6-LMW fragment possesses only a partial bZIP domain and is unable to bind the transcription factor NF-Y to form heterodimers on the ERSE motif (Yoshida et al, 2001). How then does infection with *Legionella* activate *HSPA5/BiP* transcript induction to a degree similar to that induced by the

ATF6-N fragment? Our experiments with cells depleted for ATF6 might suggest a possible mechanism (Fig 3E). In these experiments, although the DTT-mediated induction of *HSPA5/BiP* transcripts is completely suppressed, there is still a residual induction of *HSPA5/BiP* in *Legionella* infected cells (Fig 3E). We speculate that this is due to other, potentially ATF6 independent, mechanisms. *Legionella* is a pathogen that is known to target specific processes in cells via multiple redundant mechanisms (Ghosh & O'Connor, 2017; O'Connor et al, 2012). As precedents, for example, protein synthesis is inhibited by up to seven secreted *Legionella* proteins that target different host proteins involved in translation (Belyi, 2020). Another example is the host GTPase Rab1 that is distinctly modified by different post-translational modifications induced by *Legionella* proteins to regulate membrane trafficking in cells (Neunuebel & Machner, 2012). Moreover, *Legionella* also secretes proteins that modify histone acetylation and methylation or regulate RNA polymerase II function, directly resulting in gene expression changes (Schmeck et al, 2008; Rolando et al, 2013; Schuelein et al, 2018). Given that our current study characterizes one of the five *L.p.* effectors (Lpg0519) that activates the ERSE reporter in cells (Fig 5C), we postulate here that some of the other effectors unearthed from our screen might augment the activity of the ATF6-LMW by impinging directly on the ERSE dependent transcriptional induction of target genes.

As the ATF6 pathway is mainly associated with the production of chaperones, ERAD components and lipid synthesis enzymes, the pro-survival attributes associated with ATF6 activation might serve to benefit pathogen survival and replication. ATF6 is recruited to the LCV during infection and might serve to sense changes within the LCV environment (Fig S1B). An attractive model to test in future studies would be to examine whether *L.p.* uses ATF6 for lipid synthesis to actively contribute to the growing membrane of the LCV during the course of the infection. Indeed, targeting of the ATF6 branch of the UPR has been used by protozoan, bacterial, and viral pathogens alike, and has been shown to contribute to intracellular replication in each system (Jheng et al, 2010; Celli & Tsolis, 2015; Galluzzi et al, 2017). While bacteria have been shown to modulate UPR activity, it is striking how *L.p.* differentially modulates the different arms of the ER stress response pathway by inhibiting IRE1, while concomitantly, activating the ATF6 pathway (Hempstead & Isberg, 2015; Treacy-Abarca & Mukherjee, 2015).

### The evolutionary arms race between *L.p.* and its mammalian host

The primary targets of the *Legionella* infectious paradigm are freshwater protists such as amoebae (Swanson & Hammer, 2000; Boamah et al, 2017). It is well understood that over evolutionary time, horizontal gene transfer of genetic information between protists and *Legionella* have enabled this bacterium to acquire unique characteristics that target conserved pathways in both

marked with "&" indicates a higher molecular weight ATF6-FL fragment; arrows mark the ATF6-FL or processed ATF6-P or ATF6 -N fragments; "#" marks an unspecific band that is detected inconsistently by the ATF6 antibody; "*" marks the opsonization antibody (a.b.) used to coat *L.p.* before infection. **(D)** Schematic of membrane bound N-terminal GFP-tagged ATF6 (red) with relevant features highlighting S1P and S2P cleavage site sequences (blue) and mutated residues for S1P and S2P cleavage site mutations (red letters). **(E)** HEK293-FcγR cells were transfected with GFP-ATF6(WT), GFP-ATF6 R415A/R416A (GFP-ATF6 S1P mutant) and GFP-ATF6 R415A/R416A and N391F/P394L (GFP-ATF6 S1P and S2P mutant). Transfected cells were incubated with 1 mM DTT for 1 h, or infected with *L. p.* (*WT* or *ΔdotA*) for 6 h and analyzed by immunoblotting using anti-GFP and anti-GAPDH antibodies. Arrows mark ATF6-FL or the processed low molecular weight ATF6 fragment. Histograms represent the densitometric analyses showing the fold change of ATF6-FL signal remaining in treated cells relative to control cells.

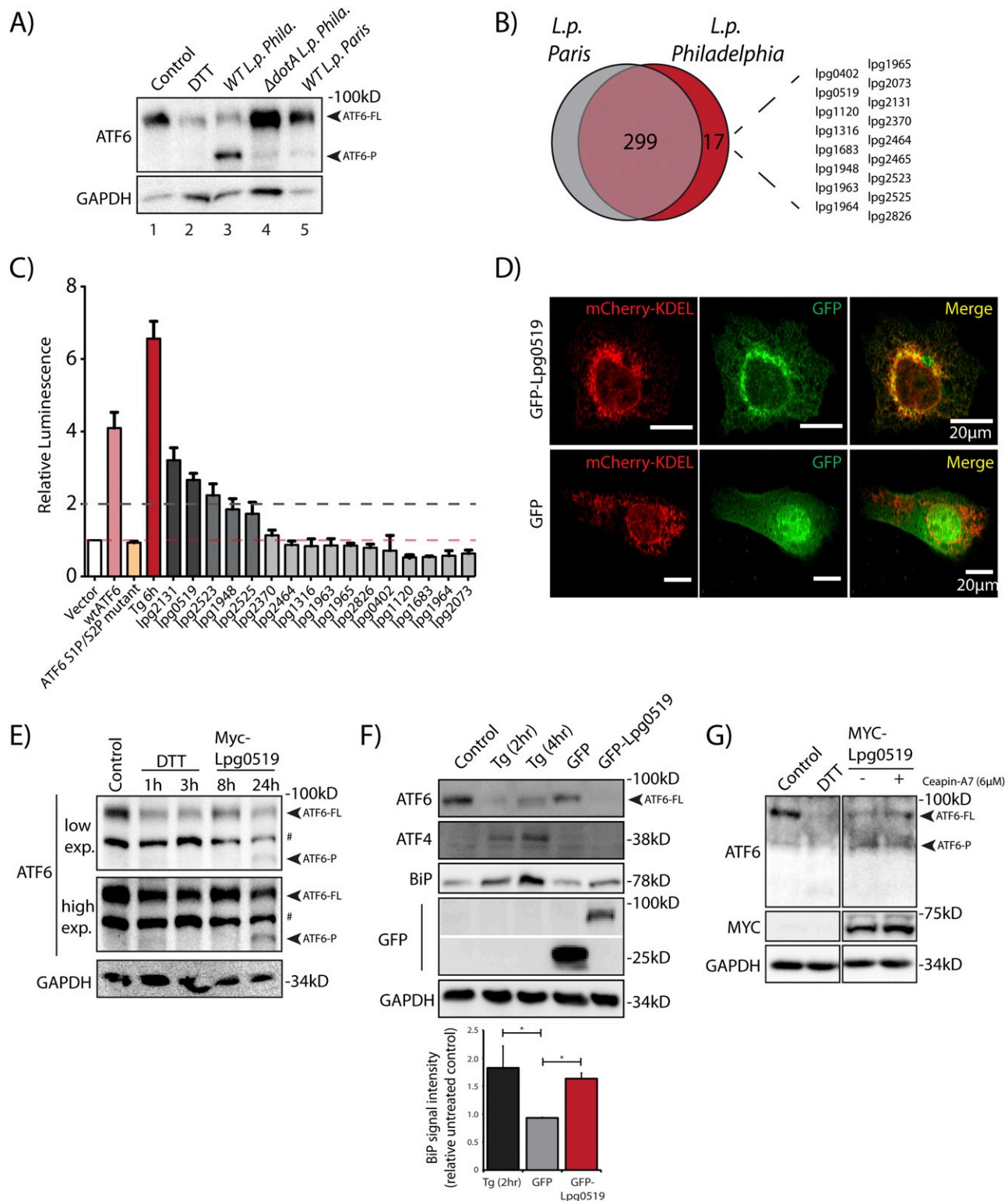

**Figure 5. Comparative genome analyses reveal *L.p.* strain specificity in processing ATF6.**
**(A)** Immunoblots of HEK293-FcγR cells cells treated with 1 mM DTT for 1 h or infected with *Legionella pneumophila* strains for 6 h–*Philadelphia* str. (*WT L.p. Phila* or Δ*dotA L.p. Phila*) and *Paris* str. (*L.p. Paris*). Lysates were immunoblotted using anti-ATF6 and anti-GAPDH antibodies. Arrow marks ATF6-FL and ATF6-P. **(B)** Identification of gene of orthologs of *L.p. Philadelphia* strain effectors through pairwise sequence alignments against *L.p. Paris* strain. Listed are *Philadelphia* strain effectors that are absent from the *Paris* strain. **(C)** Histograms depict luminescence units relative to control from the ERSE-Luciferase reporter cell line. HEK293T-ERSE-Luciferase cells were transiently transfected with Myc-tagged *Legionella* effectors or empty vector, GFP-ATF6(WT), or GFP-ATF6(R415A/R416A, N391F/P394L) (GFP-ATF6 S1P/S2P mutant). Fold induction of luciferase activity from transfected samples were calculated relative to empty control vector transfected cells. Baseline luciferase activity from control cells (red line) and

amoebae and macrophages, thus making it suitable to infect both their unicellular and mammalian hosts (Gomez-Valero et al, 2011). Examples also exist of effector proteins that elicit paradoxical responses in each of these host cells. One such an effector protein is LamA, an amylase that degrades glycogen when released into infected cells - in amoebae, LamA restricts cell wall formation and encystation by depriving cells of glycogen, thereby ensuring a permissive host for its replication. In macrophages, however, glycogen deprivation triggers innate immune responses that partially restricts bacterial replication (Price et al, 2020). In this study, we uncovered a novel, non-canonical mode of ATF6 dependent transcription that is stimulated by *Legionella* and bypasses the need for ATF6 to traffic to the Golgi and require the activity of S1P and S2P enzymes (Videos 2 and Figs 4 and S3A and B). Homologues of neither ATF6, nor S1P or S2P enzymes are present in amoebae, suggesting that this mechanism is a recent evolutionary acquisition. Even more surprising is the realization that different strains of bacteria (*Philadelphia* or *Paris*) that presumably evolved from the same species of *L. pneumophila* process ATF6 differently in their mammalian hosts due to a small but significant divergence in their effector repertoires (Burstein et al, 2016). One of these effectors, unique to *L.p. Philadelphia* is Lpg0519, a poorly studied protein, that when expressed in cells processes ATF6, activates ERSE dependent transcription, and localizes to the ER (Fig 5B–G). Bioinformatic analyses of the primary sequence of Lpg0159 suggest that this protein carries at least one transmembrane domain. However, structure prediction and homology modelling of Lpg0519 using tools such as Raptor-X, Phyre and AlphaFold unfortunately yielded us with poor prediction scores and no further insights into the mechanism of its action. Our preliminary experiments with a subset of cell permeable protease inhibitors, while effective in preventing the cleavage of DTT induced ATF6-FL (Okada et al, 2003), did not prevent *L.p.* induced ATF6-FL loss (Fig S3C and D). This suggests a role for an as yet unidentified host or *L.p.* derived protease that specifically cleaves ATF6 to generate the ATF6-P and ATF6-LMW fragments. Currently, our working model highlights two mutually exclusive hypotheses that remain to be tested: (a) Lpg0519 folds and functions as an atypical protease; or (b) Lpg0519 promotes the activity of an ER localized host protease. In either scenario, the answers to these questions using molecular tools derived from *L. pneumophila*, will enhance our understanding of ATF6 regulation in physiology and pathology.

# Materials and Methods

### Bacterial strains

All *Legionella* strains were gifts from Craig Roy's laboratory at Yale University. *Legionella* strains used in this study were routinely cultivated on Charcoal Yeast Extract agar. The Δ*dotA*, Δ*sidC-sdcA*,

and IPTG-inducible Halo-expressing strains were derived from the parental *Lp01* strain. The Δ*2,3,4,6,7*, Δ*2,3,6,7*, and Δ*2,3,6 L.p.* strains (O'Connor et al, 2012), and the Δ*7-translation L. pneumophila* strain (Barry et al, 2013) was a gift from Russell Vance's laboratory and are thymidine auxotrophs derived from the *L. pneumophila serogroup* 1 Lp02 (Berger & Isberg, 1993). *L. pneumophila Paris B1* strain was purchased from ATCC (ATCC 700833). Chloramphenicol (10 μg/ml), IPTG (0.1 mM), and thymidine (100 μg/ml) were added to Charcoal Yeast Extract agar plates as needed. *L.p.* were harvested from 2-d heavy patches and used to infect cells.

### Recombinant DNA

GFP-ATF6, HA-ATF6, and HA-ATF6 1–373 were kind gifts from Ron Prywes (plasmids #32955, #11974 and #27173; Addgene). The GFP-ATF6 truncation mutants were cloned from the GFP-ATF6 backbone and generated by Genscript Inc. For in vitro transcription and translation, the ATF6 1–331 fragment was sub-cloned between HindIII and SalI sites on the pGEM3Z vector carrying an SP6 promoter. GFP-ATF6 S1P mutant and GFP-ATF6 S1P/S2P mutant constructs were generated by site directed mutagenesis using the Q5 Site-Directed Mutagenesis Kit (NEB). For the effector screen, Myc-tagged effector proteins were amplified from a plasmid effector library that was a kind gift from Russell Vance, University of California, Berkeley. To generate N-terminal GFP tagged Lpg0519, *lpg0519* was amplified from *L.p.* (Lp01) genomic DNA and cloned into the pEGFP-C2 vector.

### Cell culture

HEK-293 FcγRII and HeLa FcγRII cells were obtained from Craig Roy's laboratory at Yale University. All cells were cultured in DMEM (Life Technologies) supplemented with 10% FBS at 37°C and 5% $CO_2$. RAW264.7 macrophages were cultured in Roswell Park Memorial Institute media (RPMI) (Corning) supplemented with 10% FBS at 37°C and 5% $CO_2$. Cells were placed in poly-L-lysine–treated plates and grown to 90% confluency. Drug treatments were performed at final concentration, 200 nM Tg (Enzo Life Sciences), 1 mM DTT (Research Products International), protease inhibitor cocktail (100 μM 4-(2-aminoethyl)benzenesulfonyl fluoride [AEBSF] [Sigma-Aldrich], 10 μM Tosyl phenylalanyl chloromethyl ketone [TPCK] [Sigma-Aldrich], 10 μM Calpain Inhibitor I [Sigma-Aldrich], 10 μM E 64 Protease inhibitor [EMD Millipore], 100 μM phenylmethylsulfonyl fluoride [PMSF] [Sigma-Aldrich]), 5 nM Bafilomycin A1 (Sigma-Aldrich), 1 μM PF-429242 (Sigma-Aldrich), 5 μM/20 μM MG-132 (Enzo Life Sciences), 25 μM Cycloheximide (Sigma-Aldrich), 10 μM Ceapin A7 (Gallagher et al, 2016). Ceapin-A7 was provided as a gift from the Peter Walter laboratory at UC San Francisco.

For transient transfections, cells were grown to 70% confluency and transfected with 2 μg of plasmid for 60 and 35 mm dishes, or 1 μg

---

twofold induction cutoff (grey line) are indicated. **(D)** Confocal micrographs from U2OS cells transfected with GFP-C2 vector (right, center) or GFP-Lpg0519 (left, center) and mCherry-Calreticulin (red, top). Scale bars = 20 μm. **(E)** HEK293T cells treated with 1 mM DTT for 1 or 3 h or transfected with Myc-Lpg0159 for 8 or 24 h and lysed. Cell lysates were subjected to immunoblotting with anti-ATF6 and anti-GAPDH antibodies. Arrows mark ATF6-FL and the processed ATF6-P fragment. "#" mark an unspecific band. **(F)** Immunoblotting of HEK293T cells treated with thapsigargin (Tg) for 2 or 4 h, or transiently transfected with GFP-C2 vector or GFP-Lpg0519. Immunoblotting was performed using antibodies against ATF6, ATF4, BiP, GFP, and GAPDH. Histograms represent quantification of replicate experiments (n = 3) of BiP signal intensity relative non-treated control cells. Mean ± SEM. *P*-values were calculated using *t* test (*$P < 0.05$). **(G)** Immunoblotting of HEK293T cells treated with Ceapin-A7 (6 μM) for 16 h followed by treatment with either 1 mM DTT for 1 h or treated with Ceapin-A7 and transfected with Myc-Lpg0159 for 16 h. Immunoblotting was performed using antibodies against ATF6, Myc and GAPDH. Arrows mark ATF6-FL and the processed ATF6-P fragment.

per well for 24-well plates using JetPRIME (Polyplus-transfection) according to the manufacturer's instructions. Cells were incubated with transfection reagent for 4 h, then media replaced with fresh DMEM supplemented with 10% FBS. For siRNA transfections, cells were grown to 30–50% confluency and transfected using Oligofectamine (Thermo Fisher Scientific) according to the manufactures protocol. Cells were grown for 72 h before application of treatment conditions. The following siRNA oligos used were purchased from Sigma-Aldrich: SEL1L-TTAACTTGAACTCCTCTCCCATAGA, Scramble-GCATACTCAACTACTTCGCATACTT; ATF6- GAACAGGGCTCAAATTCTC, Scramble-GCTAGTGCACAAGTACCTA.

### Infection

Cells were infected at a MOI of 100, 50, 25 or 5. If cells required opsonization, *Legionella* polyclonal antibody (Cat. no. PA1-7227; Invitrogen) was used at 1:2,000 and incubated for 20 min at room temperature. Immediately after infection, cells were centrifuged for 5 min at 100*g*. After centrifugation, cells were left at 37°C for an additional 60 min. After 1 hour, cells were washed with 1× PBS to remove extracellular bacteria. DMEM supplemented with 10% FBS or the same media supplemented with treatment reagent was replaced after the washes in PBS. Infected cells were harvested at the designated times.

### Quantitative RT–PCR

HEK-293 FCγ cell mRNA was harvested and isolated using Direct-zol RNA Miniprep Plus (Zymo Research) according to the manufacturer's protocol. cDNA synthesis was performed using QuantiTect Reverse Transcription Kit (QIAGEN) and cDNA reactions were primed with poly dT. Relative quantitative PCR was performed using iTaq Universal SYBR Green Supermix (Bio-Rad). GAPDH or HPRT mRNA (for human cells) or Actin mRNA (for RAW264.7) was used for normalization. Uninfected and untreated HEK-293 FCγRII and RAW264.7 macrophage cells were used as the endogenous control for each qRT-PCR analysis. The following qRT-PCR primers were used: *BiP* (Human) forward- CATCACGCCGTCCTATGTCG, reverse- CGTCAAAGACCGTGTTCTCG; *HERPUD1* (Human) forward- AACGGCATGTTTTGCATCTG, reverse- GGGGAAGAAAGGTTCCGAAG; *SEL1L* (Human) forward- AAACCAGCTTT-GACCGCCAT, reverse- GTCATAGGTTGTAGCACACCAC; *HYOU1* (Human) forward- GAGGAGGCGAGTCTGTTGG, reverse- GCACTCCAGGTTTGA-CAATGG; *ATF6* (Human) forward- AGAGAAGCCTGTCACTGGTC, reverse- TAATCGACTGCTGCTTTGCC; *DNAJB11* (human) forward- AACCTGAG-CACCTTTTGCCT, reverse- GGTTCCGGTCGGGATGAAG, *BiP* (mouse) forward- ACTTGGGGACCACCTATTCCT, reverse- GTTGCCCTGATCGTTGGCTA; *Dnajb11* (mouse) forward- TTGGAGGAACCCCTCGTCA, reverse- CTCTTGCCGA-CAGTTGCATTT; *Hsp90B1* (mouse) forward- GTTCGTCAGAGCTGATGATGAA, reverse- GCGTTTAACCCATCCAACTGAAT; *Atf6* (mouse) forward-TCGCCTTTTAGTCCGGTTCTT, reverse- GGCTCCATAGGTCTGACTCC; *Actin* (mouse) forward- GGCTGTATTCCCCTCCATCG, reverse- CCAGTTGGTAA-CAATGCCATGT.

### Immunoblotting analysis

Mammalian cells were lysed in radioimmunoprecipitation assay buffer (RIPA) buffer with the addition of protease (Roche cOmplete), and phosphatase inhibitors (GB Sciences). Protein levels of lysates were determined using the Bio-Rad DC/RC assay. Equal amounts of protein lysate were boiled with SDS load buffer, and equal amounts of protein were loaded. Immunoblotting was performed with the following antibodies: GAPDH (Cat. no. 60004-1-lg; Proteintech), ATF6 rabbit polyclonal (Cat. no. 24169-1-AP; Proteintech), ATF6 mouse monoclonal (Cat. no. 66563-1-Ig; Proteintech), *β*-Actin (Cat. no. 20536-1-AP; Proteintech), *α*-Tubulin (Cat. no. 66031-1-Ig; Proteintech), SEL1L (Cat. no. ab78298; Abcam), ABCD3 (Cat. no. 66697-1-Ig; Proteintech), GFP tag (Cat. no. 66002-1-Ig; Proteintech), BiP (Cat. no. 11587-1-AP; Proteintech), ATF4 (Cat. no. 11815S; Cell Signaling Technology).

### Immunofluorescence microscopy

Cells were plated on 12 mm glass coverslips in 24-well plates. After 24 h, cells were fixed, treated with drugs, or infected with *L.p.* as needed. For ubiquitin recruitment assays, HEK293 FcγR cells were infected with *L.p.* at MOI = 5. 1 h after infection, cells were washed twice with PBS to remove extracellular bacteria and incubated for 2 h more. For co-localization assays, Cos7 cells were co-transfected with pcDNA-FcγRII and GFP-ATF6*α*, then infected with *L.p.* at MOI = 10 and infected for 1, 4, or 8 h. For 4- and 8-h time points, cells were washed with PBS after 1 h to remove extracellular bacteria. Coverslips were mounted with ProLong Diamond Antifade Mountant (Thermo Fisher Scientific) incubated at 37°C for 10 min, then imaged directly. For immunofluorescence, coverslips were washed with cold 1× PBS, fixed with 4% paraformaldehyde in PBS for 10 min at room temperature, permeabilized in 0.1% saponin in PBS, and blocked in 3% BSA in PBS, and then incubated with the appropriate primary and secondary antibodies diluted in 3% BSA. Nuclei were stained with Hoechst 33342 dye for 10 min before mounting on microscope slides. Coverslips were imaged using an inverted Nikon Eclipse Ti-E spinning disk confocal microscope equipped with a Prime 95B 25mm CMOS camera (Photometrics) camera. Antibodies used were Ubiquitin (Cat. no. ST1200-100UG; Millipore), and Secondary Antibody, Alexa Fluor 546 or Alexa Fluor 488 (Thermo Fisher Scientific).

### Live cell microscopy

HeLa FcγRII cells expressing GFP-ATF6 were plated on 35 mm poly-lysine–coated imaging dishes (Cellvis). Cells were infected at MOI = 5 with *WT*- or Δ*dotA L.p.* that were previously stained with HaloTag-Janelia Fluor 646 conjugates (Grimm et al, 2017). For staining, briefly, *L.p.* maintaining a HaloTag-expressing plasmid were harvested from a 2-d heavy patch and incubated at liquid culture overnight with 0.1 mM IPTG. Liquid culture at OD = 3.0 was then pelleted and resuspended in 5 mM Janelia Fluor 646 HaloTag ligand (JF646) to facilitate HaloTag-JF646 conjugation. After 15 min incubation with ligand in the dark, *L. pneumophila* were washed 1× with water. Stained bacteria were resuspended in 2 ml of DMEM lacking phenol red (Gibco) and used for infection of cells. Imaging of cells took place in a controlled chamber maintaining 37°C with 5% $CO_2$. A random selection of cells was imaged at 60× magnification at 5- or 10-min intervals for 12 h using a Nikon Eclipse Ti2 microscope with a Nikon DS-Qi2 camera. Cells that died or lost focus over the time course were omitted from analysis.

## In vitro transcription and translation

In vitro transcription and translation of luciferase and the ATF6 1–331 fragment were performed using the TnT coupled Reticulocyte Lysate Systems kit from Promega. Briefly, 1 µg of DNA encoding for luciferase or the ATF6 1–331 fragment downstream of an SP6 promoter were incubated with TnT reaction buffer, RNA polymerase, amino acids, RNAsin ribonuclease inhibitor (Promega), Magnesium actetate (0.25 mM) and rabbit reticulocyte lysate as a source of ribosomes and translation machinery in a test tube at 30°C for 90 min. 20% of the total reaction mixture was then boiled with Lammeli buffer and run on a 4–15% reducing SDS–PAGE gel before immunoblotting with an ATF6 antibody (rabbit; Proteintech).

## Luciferase assay

ONE-Glo Luciferase assay system was purchased from Promega. HEK-293T ERSE-Luciferase cells, provided as a gift from the Peter Walter laboratory at UC San Francisco and described previously (Gallagher et al, 2016), were seeded onto six-well dishes at 70% confluency. Cells were transfected with either GFP- or HA-tagged ATF6 constructs for 48 h or Myc-tagged *Legionella* effectors as previously described (Barry et al, 2013) for 24 h. Cells were suspended in DMEM and 50 µL aliquots were transferred to assay plate (Cat. no. 353296; Falcon) in triplicate. One-Glo Luciferase reagent was pre-equilibrated to room temperature and added at equal volume into each well. Assay plate measurements were performed on Tecan Saphire[2] with 1 s exposure. Background signal was subtracted from untreated untransfected cells. Treatment conditions were normalized to control cells and represented as fold change in relative luminiscence.

## Quantification statistical analysis

GraphPad Prism 6 software was used for statistical analysis. Where statistical analysis was performed an unpaired *t* test was performed using three biological replicates. Statistical significance: *$P$-value < 0.05; **$P$-value<0.01, ***$P$-value <0.001. Image analysis was performed using ImageJ.

# Supplementary Information

# Acknowledgements

We thank Peter Walter for scientific advice and for very generously providing us with the ERSE cell-line and Ceapin A7 inhibitor. We thank Dr. Philipp Schlaermann for doing preliminary studies on ATF6. We would like to acknowledge Dr. Ralph Isberg (Tufts University) and Dr. Russell Vance (UC, Berkeley) for providing us with the Myc-tagged *Legionella* effector library and Dr. Craig Roy (Yale University) for providing us various *Legionella* strains and species. We thank all members of the Mukherjee lab for critical reading of the manuscript. S Mukherjee is supported by the National Institutes of Health RO1 grant AI118974 and an award from the Pew Charitable Trust (A129837).

## Author Contributions

NU Ibe: conceptualization, data curation, formal analysis, investigation, visualization, methodology, and writing—original draft.
A Subramanian: conceptualization, data curation, formal analysis, validation, investigation, visualization, methodology, and writing—review and editing.
S Mukherjee: conceptualization, resources, supervision, funding acquisition, visualization, project administration, and writing—original draft, review, and editing.

## Conflict of Interest Statement

The authors declare that they have no conflict of interest.

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
