## [Reviewer comments · Life Science Alliance]

Life Science Alliance

Non-canonical activation of the ER stress sensor ATF6 by *Legionella pneumophila* effectors

Nnejiuwa Ibe, Advait Subramanian, and Shaeri Mukherjee

DOI: <https://doi.org/10.26508/lsa.202101247>

Corresponding author(s): Shaeri Mukherjee, University of California, San Francisco

Review Timeline:	Submission Date:	2021-09-27
	Editorial Decision:	2021-09-29
	Revision Received:	2021-09-29
	Accepted:	2021-10-01

Transaction Report:

Please note that the manuscript was previously reviewed at another journal and the reports were taken into account in the decision-making process at Life Science Alliance.

September 29, 2021

RE: Life Science Alliance Manuscript #LSA-2021-01247-T

Dr. Shaeri Mukherjee
UCSF
513 Parnassus Avenue
HSW 1522
San Francisco, CA 94143

Dear Dr. Mukherjee,

Thank you for submitting your revised manuscript entitled "Non-canonical activation of the ER stress sensor ATF6 by Legionella pneumophila effectors". We would be happy to publish your paper in Life Science Alliance pending final revisions necessary to meet our formatting guidelines.

- please add ORCID ID for the corresponding author-you should have received instructions on how to do so
- please add a Summary Blurb/Alternate Abstract in our system
- please add a Category for your manuscript in our system
- please add the Twitter handle of your host institute/organization as well as your own or/and one of the authors in our system
- please add the contribution of all Authors to our system

A. FINAL FILES:

B. MANUSCRIPT ORGANIZATION AND FORMATTING:

Sincerely,

October 1, 2021

RE: Life Science Alliance Manuscript #LSA-2021-01247-TR

Dr. Shaeri Mukherjee
University of California, San Francisco
513 Parnassus Avenue
HSW 1522
San Francisco, CA 94143

Dear Dr. Mukherjee,

Thank you for submitting your Research Article entitled "Non-canonical activation of the ER stress sensor ATF6 by Legionella pneumophila effectors". It is a pleasure to let you know that your manuscript is now accepted for publication in Life Science Alliance. Congratulations on this interesting work.

DISTRIBUTION OF MATERIALS:

Again, congratulations on a very nice paper. I hope you found the review process to be constructive and are pleased with how the manuscript was handled editorially. We look forward to future exciting submissions from your lab.

Sincerely,
